# Firing Performance of Microchip Exploding Foil Initiator Triggered by Metal-Oxide-Semiconductor Controlled Thyristor

**DOI:** 10.3390/mi11060550

**Published:** 2020-05-29

**Authors:** Ke Wang, Peng Zhu, Cong Xu, Qiu Zhang, Zhi Yang, Ruiqi Shen

**Affiliations:** 1School of Chemical Engineering, Nanjing University of Science and Technology, Nanjing 210094, China; 117103010448@njust.edu.cn (K.W.); congxu@njust.edu.cn (C.X.); qiuzhang@njust.edu.cn (Q.Z.); yangzhi@njust.edu.cn (Z.Y.); rqshen@njust.edu.cn (R.S.); 2Micro-Nano Energetic Devices Key Laboratory, Ministry of Industry and Information Technology, Nanjing 210094, China

**Keywords:** microchip exploding foil initiator, metal-oxide-semiconductor controlled thyristor, capacitor discharge unit, micro-electro-mechanical system

## Abstract

In this paper, microchip exploding foil initiators were fabricated by micro-electro-mechanical system scale fabrication methods, such as magnetron sputtering, photolithography, and chemical vapor deposition. A small-scale capacitor discharge unit based on the metal-oxide-semiconductor controlled thyristor was designed and produced to study the performance of the microchip exploding foil initiator. The discharge performance of the capacitor discharge unit without load and the effect of protection devices on the metal-oxide-semiconductor controlled thyristor were studied by the short-circuit discharge test. Then, the electric explosion characteristic of the microchip exploding foil initiator was also conducted to study the circuit current, peak power, deposited energy, and other parameters. Hexanitrostilbene refined by ball-milling and microfluidic technology was adopted to verify the initiation capability of the microchip exploding foil initiator triggered by the metal-oxide-semiconductor controlled thyristor. The results showed that the average inductance and resistance of the capacitor discharge circuit were 22.07 nH and 72.55 mΩ, respectively. The circuit peak current reached 1.96 kA with a rise time of 143.96 ns at 1200 V/0.22 μF. Hexanitrostilbene fabricated by ball-milling and microfluidic technology was successfully initiated at 1200 V/0.22 μF and 1100 V/0.22 μF, respectively.

## 1. Introduction

With an increasingly harsh operating environment, higher requirements are placed on the safety and reliability of pyrotechnic devices. The exploding foil initiator (EFI) has more advantages over traditional ignition and initiation devices on reliability and safety, since there are no primary explosives in the initiation sequence [1,2,3,4]. The EFI, as a part of the pulse power unit or capacitor discharge unit (CDU), plays an important role in electronic safety and arming device (ESAD) [5]. Development requirements for miniaturization, low energy, and integration of the EFI have promoted the application of advanced technology in the preparation of the EFI. From the early assembly of discrete components to the use of silicon integrated processes, then to low temperature co-fired ceramic (LTCC) technology and the micro-electro-mechanical system (MEMS) technology, the application and improvement of new technologies have greatly improved the working performance of the EFI [6,7,8]. Specifically, the MEMS technology, which is derived from the semiconductor process, has been used widely in the fabrication of the microchip exploding foil initiator (McEFI) since the introduction of the concept of chip-based pyrotechnic devices [9,10,11]. The MEMS technology can be applied in film preparation, micro-nano structure manufacturing of pyrotechnic devices, and the integration of energetic microsystems.

The firing capability of the McEFI is greatly influenced by the performance of the CDU. A CDU consists of a high-voltage capacitor, high-voltage switch, and exploding foil initiator. The high-voltage switch controls the flow of energy from the capacitor to the exploding foil, which makes a great difference in the performance of the EFI. The reason is that its inductance and resistance affect the rise time of the circuit current and peak current seriously, according to the resistance-loop-capacitance (RLC) circuit theory. Nowadays, the common high-voltage switches used in the EFI system mainly include gas switch (cold cathode trigger tube (CCT) and vacuum switch), planer triggered spark gap switch (PTS), semiconductor switch (metal-oxide-semiconductor controlled thyristor (MCT)), and single shot switch (or planer dielectric switch) [12,13,14,15,16,17,18]. The advantages of the commercial switch include higher withstand voltage and maturity. The planer spark gap switch derives from the gas switch, and it is easy to integrate with the EFI and has a smaller size. The single shot switch has received more attention because of its planarity, integration, and low cost. Compared with other switches, the advantages of the semiconductor switch, especially the MCT switch, include stability, reusability, and easy triggering. Firstly, the low voltage trigger mode of the MCT is different with the high voltage trigger mode of other switches, which can make the power supply structure simplified and improve the system safety. Secondly, the MCT of the solid medium has higher stability and repeatability than the gas medium switch like CCT and PTS, because the solid medium is hardly affected by environment temperature and pressure. Besides, the use of the semiconductor switch is in line with the development trend of intelligent and digital pyrotechnic devices. In addition, the withstand voltage of the MCT is lower than that of the gas switch, and planer triggered spark gap switch and the MCT have no minimum operating voltage limits like gas switch, so it is suitable for low energy initiation of the EFI. However, most research is concerned with the discharge characteristics of the MCT switch, and no public report focuses on the use of the MCT switch to control the initiation of the McEFI [12,16]. Based on our research, this study improved the triggering and protection methods of the MCT, optimized the circuit configuration of the CDU, and eventually achieved the initiation of the McEFI.

In this paper, firstly, the McEFI was designed and fabricated by MEMS technology. In order to research the electric explosion performance of the McEFI, the MCT switch and high-voltage ceramic capacitor were adopted to make up a CDU. The inductance and resistance of the discharge circuit was measured by the short-circuit discharge test. In addition, the effect of protection devices on the safety of MCT was also studied in the short-circuit test. The electric explosion parameters of McEFI was obtained at different capacitor voltages. The McEFI with two kinds of the hexanitrostilbene (HNS) was used to do the initiation test. The purpose of this paper was to study the electrical and initiation performance of McEFI when the MCT acted as the high-voltage switch.

## 2. Materials and Methods

The MEMS technology has higher machining accuracy than the conventional process, which can precisely control the parameters of the components including exploding foil, flyer, and barrel, thus reducing the initiation energy. Besides, the integrated preparation of the EFI using the MEMS replaces the manual assembly of the EFI components, also reducing the assembly error. Furthermore, the MEMS can achieve mass production of the EFI, which reduces costs and then expands the application area.

The McEFI was manufactured by MEMS scale fabrication methods, including magnetron sputtering, photolithography, and chemical vapor deposition. The McEFI contains the following components, ceramic substrate, copper foil or exploding foil, mixed flyer, SU-8 barrel, and the HNS pellet. The schematic diagram of the McEFI preparation process and the cross-section of the single McEFI are shown in Figure 1. Firstly, the 3.6 μm copper film was deposited onto a 2.5 × 2.5 inch chemical cleaned ceramic substrate by magnetron sputtering. Subsequently, the exploding foil was formed by using the photolithography to etch excess copper. The size of the bridge area was 0.4 × 0.4 mm. Then, the samples were transferred into a chemical vacuum deposit (CVD) chamber to deposit a 25 μm parylene C (PC) layer. After that, the samples were magnetron sputtered one more time to form a 2 μm Cu layer on the PC layer. The Cu layer and PC layer together acted as the mixed flyer to impact the explosive. The purpose of the Cu layer was to conveniently measure the speed of the mixed flyer, avoiding the difficulty of collecting the flyer speed due to the light transmittance of the PC layer. In addition, it was found in the experiment that the PC-Cu mixed flyer had a better initiation effect than the pure PC flyer. The 400 μm high barrel was made of SU-8 epoxy negative photoresist to shear and accelerate the flyer to a high speed. The diameter of the barrel was 0.6 mm. The samples of the McEFI after scribing are shown in Figure 2. The size of a single McEFI is 5 mm (*L*) × 3.1 mm (*W*) × 0.635 mm (*H*). The working mechanism of the McEFI is that the static energy stored in the high-voltage capacitor is quickly transferred to the bridge area of the exploding foil through the transmission line under the control of the high-voltage switch, and then, the fast energy deposition makes the bridge convert into a high pressure and temperature plasma. The plasma shears and accelerates the PC-Cu flyer to a high speed along the Su-8 barrel [19,20]. Finally, the flyer impacts the HNS pellet and the explosive is initiated due to the super pressure generated by this impact [21,22,23].

The schematic diagram of the CDU based on the MCT switch is shown in Figure 3. The MCT switch is a solid-state semiconductor high-voltage switch, which is turned on and off by controlling the voltage of the gate (G) and the gate return (GR). The drive and protection of the MCT need to be considered in the design of the CDU. For the pulse power system like the EFI system, it is required that the energy stored in the capacitor should be transferred to the bridge foil as fast as possible, which means the conduction delay of the MCT should be as short as possible. Therefore, the drive chip was adopted to assure the quick conduction of the circuit. In the process of the capacitor discharge, the inductance between the G and GR can lead to induced voltage beyond the maximum limit under the high current. The transient voltage suppressor (TVS) between the G-GR can clamp the G-GR voltage under the maximum value and avoid the breakdown of the G-GR. In addition, the TVS also acted as an isolation device between the control signal circuit and main discharge circuit, reducing or prohibiting the interference of the latter to the former. Due to the unidirectional conduction of the MCT, the fast recovery diode (FRD) in parallel with the switch can provide the reverse current channel and avoid the reverse current shock to MCT. In the experiment, the loop resistance R_0_ and circuit parasitic inductance L were studied in the short-circuit (without McEFI) discharge test, as well as the influence of the TVS on the G-GR induced voltage. In the electric explosion test, the voltage across the McEFI and circuit current at different capacitor voltages was recorded by digital oscilloscope.

As shown in Figure 3, the FRD can be connected in parallel with the capacitor (①) or MCT (②). Especially when the loop reverse current’s last time and value is bigger, the second connection mode can increase the energy absorption of the EFI or capacitor energy utilization, because the reverse current still flows through the McEFI. The second connection mode was adopted in the experiment. The L and R_0_ were the parasitic inductance and equivalent resistance of the discharge circuit, which influence the rise time of the circuit discharge current and maximum.

The refined HNS was used to verify the initiation ability of the McEFI. The HNS is regarded as the standard explosives in the in-line detonator because of its physical and chemical stability and short-pulse sensitivity [24,25]. The HNS sensitivity is mostly decided by its particle size and particle size distribution. Therefore, the HNS refined by both microfluidic technology and ball-milling were adopted to verify the initiation capability of the CDU and McEFI, and their mean particle sizes (D50) measured by laser particle size analysis were 373 and 158 nm, respectively. The HNS refined by ball-milling was obtained by the ball crusher, which is a purely physical grinding process. The microfluidic technology prepared the refined HNS by recrystallizing HNS-II. Compared with the HNS made by ball-milling, the HNS refined by microfluidic technology has smaller particle size and narrower particle size distribution range [26]. In the initiation test, the HNS pellet was fixed on the SU-8 barrel with tape. Then, the CDU was placed in an explosion-proof box and connected with the pulse source outside by charging cable.

## 3. Results and Discussions

### 3.1. Short-Circuit Discharge Test

The purpose of the short-circuit discharge test was to verify whether the circuit had the ability to produce the pulse big current. The indicator for measuring the qualification of the discharge circuit for the EFI application is that the short-circuit current curve has five positive peaks at least. Figure 4 illustrates the discharge curves of the circuit at multiple capacitor voltages, and it can be clearly seen that the current curve includes at least six obvious peaks. According to the R-L-C circuit Equations (1)–(3), the parasitic inductance L and circuit resistance *R*_0_ can be obtained [27]:(1)$$T=\frac{2\pi }{\omega }=\frac{2\pi }{\sqrt{1/LC-R_{0}^{2}/4L^{2}}},$$
(2)$$L=\frac{T^{2}}{C}\left(4\pi^{2}+{\left(\ln I_{1}-\ln I_{2}\right)}^{2}\right){)}^{-1},$$
(3)$$R_{0}=\frac{2L}{T}\ln\frac{I_{1}}{I_{2}},$$
where *T*, *C*, *I*_1_, *I*_2_ represent the current period, capacitance, first peak current, and second peak current. The period *T* is the time difference between the first peak current and second peak current. Table 1 shows the related short-circuit discharge parameters and *L* & *R*_0_ calculated.

The values of *R*_0_ and *L* at different voltages are not equal, especially the *R*_0_. This may be due to the resistance change of the FRD at different capacitor voltages. The average *L* and *R*_0_ calculated by the above parameters are 72.55 mΩ and 22.07 nH. Strictly speaking, the above equations to calculate *L* and *R*_0_ were not very reasonable, because the positive current loop and the reverse current loop were not completely consistent. The unidirectional conductivity of the MCT caused the reverse current to flow through the FRD, which introduced interference into the calculation. The simulated short-circuit discharge curve and the actual discharge curve at 1200 V/0.22 μF are displayed in Figure 5. The simulated current curve is based on Equations (4)–(6):(4)$$i\left(t\right)=C\frac{dU_{c}}{dt}=-\frac{U_{0}}{wL}e^{-\delta t}sinwt,$$
(5)$$\delta =\frac{R}{2L},$$
(6)$$w=\sqrt{w_{0}{}^{2}-\delta^{2}}=\sqrt{\frac{1}{LC}-{\left(\frac{R}{2L}\right)}^{2}}.$$

The change trend of the simulated curve is basically consistent with the actual curve before 1.25 μs, but there still exists a certain deviation. This is caused by two reasons mainly. One is the calculation error of the parameter, and the other is that the simulated curve is ideal, with no consideration of the FRD.

The gate(G)-gate return (GR) peak voltage at multiple capacitor voltages was collected in Table 2.

The voltage limit of gate voltage (*Vg*) of the MCT is between +25 V and −25 V. It is evident that the TVS effectively protects the MCT from excessive induced voltage. The curves of *Vg*, the circuit current, and capacitor voltage in the short-circuit process are shown in Figure 6.

When the *Vg* reaches the trigger inflection point (*Vg* = 6.2 V), the MCT switch begins to conduct and the circuit current begins to increase. The time difference ΔT from the trigger initial point to the current initial point, which is the delay time of the MCT switch, is approximately equal to 160 ns. Then, the Vg decreases because of the large current change rate *di/dt* at the initial moment. After the trigger inflection point, the curves of *Vg* and the current are symmetrical, or the change trend of them is opposite. This is because the value of *Vg* controls the conduction degree of the switch. When the loop current increases, the inductance causes Vg to decrease to suppress the current increase. When the loop current decreases, the inductance causes Vg to increase to suppress the current decrease. The oscillation amplitude of Vg decreases as the discharge current decreases.

Compared with reference [14], it is clear that there exists a distance between the trigger inflection point and the current initial point for the planer trigger switch. A similar situation will also appear in other gas medium switches. This time distance is the process of breakdown of the gas medium, the generation of carriers, and the establishment of a conduction path. However, this situation does not happen in the MCT switch, because the existed carriers will quickly conduct under the action of the electric field, thereby reducing the delay time.

### 3.2. Electric Explosion Test

Compared with other high-voltage switches used in the EFI, the lower withstand voltage of the MCT is conductive to study the metal electric explosion and initiation of the EFI at lower input energy, which has become the application advantage of the MCT in the low energy EFI (LEEFI). The EFI is a device that converts electrical energy into flyer kinetic energy and relies on the high-speed flyer to impact the explosive to achieve initiation. Therefore, the explosion parameters of the EFI can be seen as the initiation capability of the EFI indirectly, especially the circuit peak current and peak power of the EFI. Figure 7 depicts the electric explosion curves of the McEFI at different capacitor voltages. Table 3 lists the related electric explosion parameters of the McEFI.

As the capacitor voltage increases, the peak current *I*_peak_, peak voltage *V*_peak_, peak power *P*_peak_, and *E_d_* increase accordingly. The *t*-*V*_peak_, which is thought as the electric explosion moment, is close to *t*-*P*_peak_. However, at 600 and 800 V, the peak voltage *V*_peak_ is much smaller than that of 1000 and 1200 V, and there is no distinct voltage spike. This indicates that the bridge foil explosion is quite incomplete or no electric explosion because of the lower input energy or smaller current, which leads to the relatively lower resistance and slower energy deposition rate of the bridge foil. Therefore, the reverse current occurs. At 1000 and 1200 V, the electric explosion is evident and the peak voltage increases significantly.

The *t*-*V*_peak_ is ahead of *t*-*I*_peak_ at 600 V and 800 V, but it lags behind *t*-*I*_peak_ at 1000 and 1200 V. It is known that the bridge resistance increases with the current before the electric explosion moment (*t*-*V*_peak_), and the phase change also happens. At 600 and 800 V, when the bridge reaches the phase state corresponding to the *t*-*V*_peak_ (solid or solid-liquid coexistence state), the resistance of the bridge foil is the largest at this time, but the initial energy stored in the capacitor is too low, so the subsequent energy input cannot support further phase state change of the bridge foil. On the contrary, at 1000 and 1200 V, the bridge foil can absorb more energy to make the metal rapidly vaporized, causing the electric explosion. The time point of the electric explosion will also advance with the increase of the initial energy.

The *E_d_* in Table 3 is the energy deposited by the McEFI up to *t*-*P*_peak_, which is generally recognized as the effective energy to push the flyer. However, since it is difficult to unify the flyer velocity curve and the electric explosion curve, the E_d_ here may be not reasonable, just as a reference. The *E_d_* at 600 and 800 V also can prove that the electric explosion did not happen. Assuming that the bridge foil does not deform significantly at the moment of the peak current, the maximum current density at 800 and 1000 V is 9.375 × 10^7^ A/cm^2^ and 1.167 × 10^8^ A/cm^2^. This also verifies that the critical current density at which the bridge foil can explode is likely 10^8^ A/cm^2^.

### 3.3. Initiation Test

In the experiment, two kinds of the HNS were selected to analyze the initiation ability of the McEFI. Their particle sizes (D50) were respectively 373 and 158 nm. The first kind (373 nm) was prepared by ball-milling, and the other (158 nm) was made by microfluidic technology. Both were pressed into the column and then fixed on the barrel of the McEFI with the tape. The pellet’s density was 1.60 g/cm^3^, with a size of 4 mm (*Φ*) × 3 mm (*H*). Figure 8 displays the firing device the CDU used to initiate the HNS. The initiation results are shown in Table 4. In the test, the minimum detonator voltage of two kinds of HNS was obtained.

As seen from Table 4, the minimum initiation energy for 373 nm HNS and 158 nm HNS is 158.4 and 133.1 mJ, which indicates that the decrease of particle size can indeed reduce initiation energy to some extent.

The initiation energy of the McEFI not only depends on the particle size of the HNS, but also the flyer speed. Flyer kinetic energy or speed is the most important indicator to assess whether the HNS can be initiated [28]. According to the P-τ theory, the dynamic pulse high pressure generated by the flyer impacting the HNS leads to the initiation of the HNS [29,30]. Therefore, the flyer speed at 1100 V/0.22 μF and 1200 V/0.22 μF were measured by photo doppler velocity (PDV) measurement system, as shown in Figure 9. The principle of PDV measurement is the doppler frequency shift effect. Two lasers emitted by the same laser have the same initial frequency. One serves as a contrast, and the other shines on the surface of the moving object and returns. The frequency difference between them can be used to calculate the speed of the mixed flyer. The curves of the flyer velocity and the moving distance with time were obtained by processing the original velocity curve measured by PDV, as shown in Figure 10.

As shown in the Figure 9 and Figure 10, the maximum flyer speed at 1100 and 1200 V reaches 2089 and 2422 m/s, respectively. When the moving distance of the flyer is equal to the height of the barrel, which is 0.4 mm high, the speed of the flyer is approximately close to the maximum velocity. The moving time of the flyer in the barrel exceeds 200 ns. The moments of maximum acceleration on the two speed curves are 53.64 and 32.65 ns, which indicates that the electric explosion plasma has the maximum pressure at that point. At the beginning of the flyer movement, the plasma pressure gradually increases because the plasma is still absorbing energy. When the pressure reaches the maximum, although the plasma is still absorbing energy, the plasma expansion is intensified due to the fast movement of the flyer, resulting in the decrease in pressure. The maximum acceleration and velocity increase with the initial capacitor voltage, because the plasma absorbs more energy in higher voltage, as shown in Table 3. Assuming that no ablation occurs in the movement of the flyer, the maximum plasma pressure can be calculated according to the Equations (7) and (8):(7)F=PS=ma,
(8)$$m=m_{Cu}+m_{PC},$$
where the *F*, *P*, *S*, and *m* are the driving strength, plasma pressure, flyer area, and the mixed flyer mass, respectively. As a result, the maximum plasma pressure reaches 1.92 and 2.67 GPa at 1100 V/0.22 μF and 1200 V/0.22 μF.

## 4. Conclusions

In this paper, a CDU based on the MCT switch was designed and fabricated. The average inductance and resistance of the CDU were 22.07 nH and 72.55 mΩ, respectively. The electric explosion test of the McEFI was conducted from 600 to 1200 V, and the circuit peak current reached 1.96 kA with a rise time of 143.96 ns at 1200 V/0.22 μF. Two types of McEFI with different particle sizes of HNS explosives were successfully initiated at 1100 V/0.22 μF and 1200 V/0.22 μF, which verified that the MCT switch had the sufficient capability for the inline initiation system. However, the relatively low withstand voltage limits the use of the MCT in higher power electric gun devices. In addition, the safety of the MCT in a complex electromagnetic environment should also be studied in future research.

## Figures and Tables

**Figure 1 micromachines-11-00550-f001:**
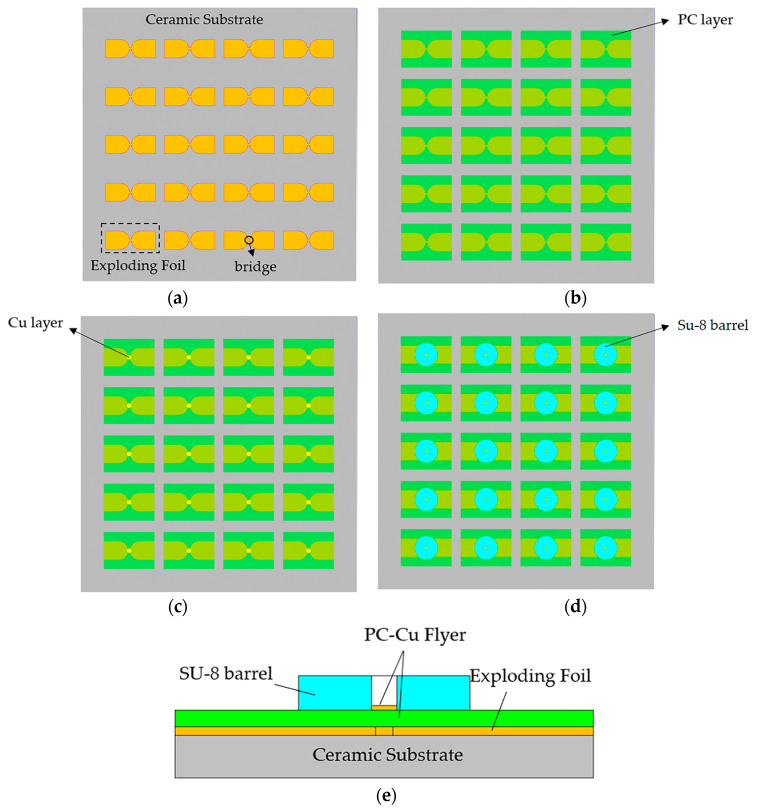
Diagram schematic of the McEFI preparation process: (**a**) exploding foil formed after removing excess copper; (**b**) PC layer deposited on the exploding foil; (**c**) Cu layer deposited on the PC; (**d**) Su-8 barrel formed by ultraviolet lithography; (**e**) cross-section schematic of a single McEFI.

**Figure 2 micromachines-11-00550-f002:**
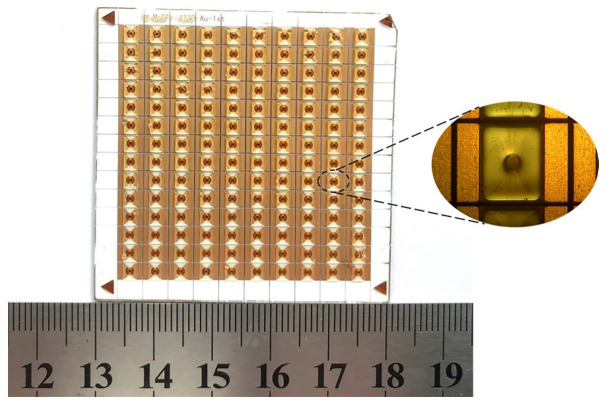
Samples of the McEFI after scribing.

**Figure 3 micromachines-11-00550-f003:**
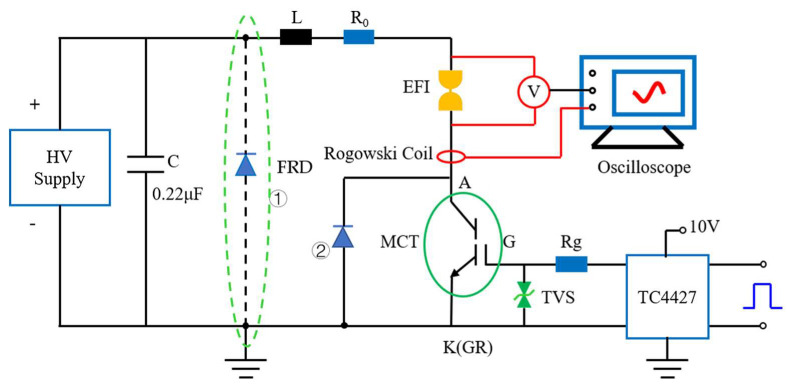
Schematic diagram of the CDU used to do the electric performance test and initiation test.

**Figure 4 micromachines-11-00550-f004:**
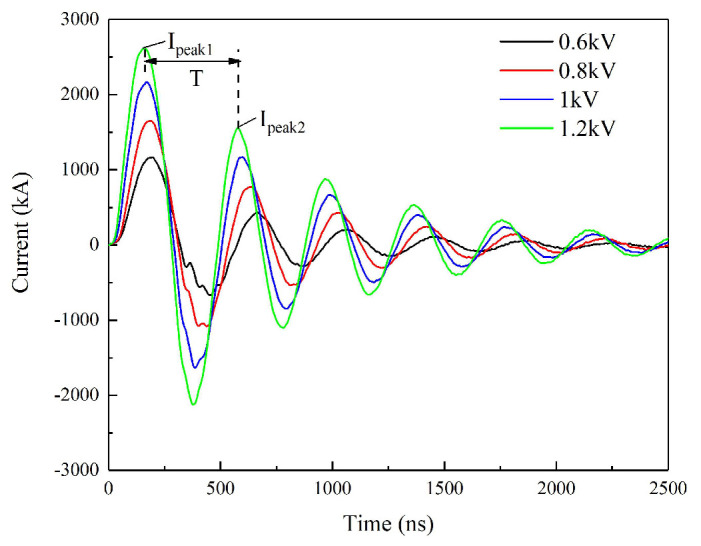
Short-circuit discharge curves at different capacitor voltages.

**Figure 5 micromachines-11-00550-f005:**
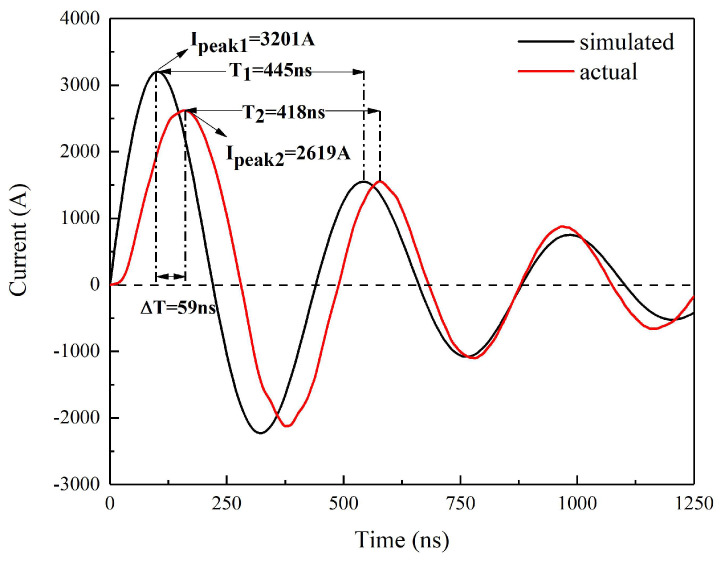
Simulated and actual short-circuit discharge curves at 1200 V/0.22 μF.

**Figure 6 micromachines-11-00550-f006:**
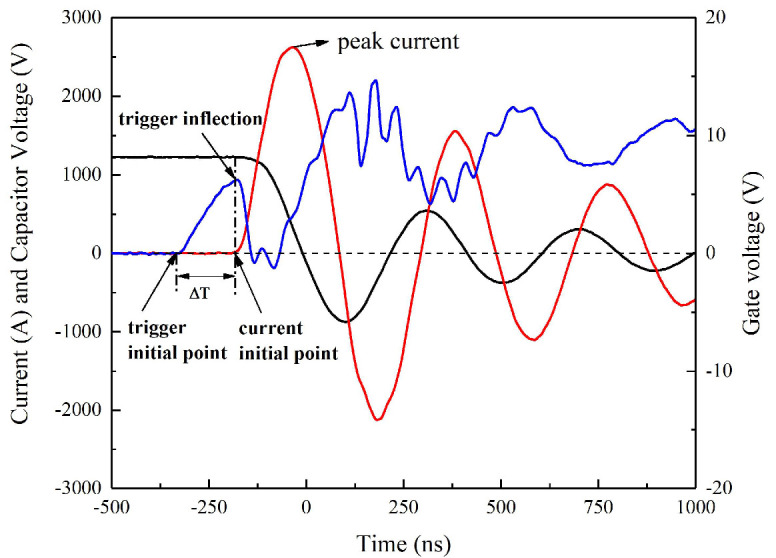
Curves of gate voltage, short-circuit, and capacitor at 1200 V/0.22 μF.

**Figure 7 micromachines-11-00550-f007:**
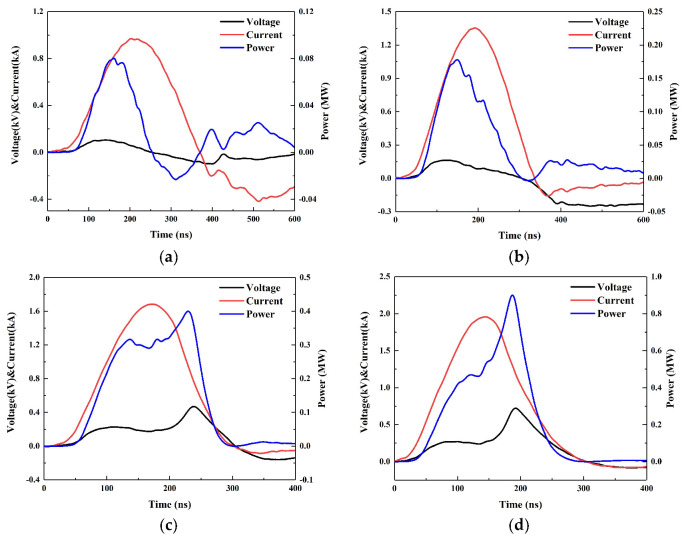
Electric explosion curves of the McEFI at different capacitor voltages (**a**) 600 V/0.22 μF; (**b**) 800 V/0.22 μF; (**c**) 1000 V/0.22 μF; and (**d**) 1200 V/0.22 μF.

**Figure 8 micromachines-11-00550-f008:**
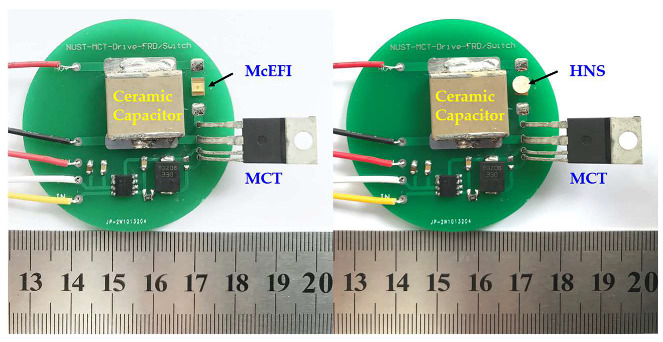
Firing device the CDU for initiating the HNS.

**Figure 9 micromachines-11-00550-f009:**
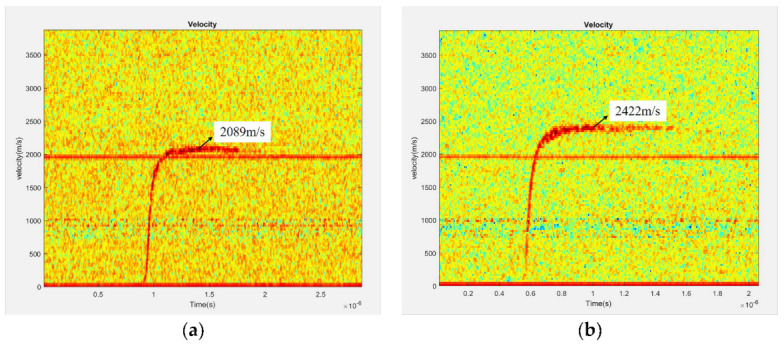
Flyer velocity curves measured by PDV at (**a**) 1100 V/0.22 μF; (**b**) 1200 V/0.22 μF.

**Figure 10 micromachines-11-00550-f010:**
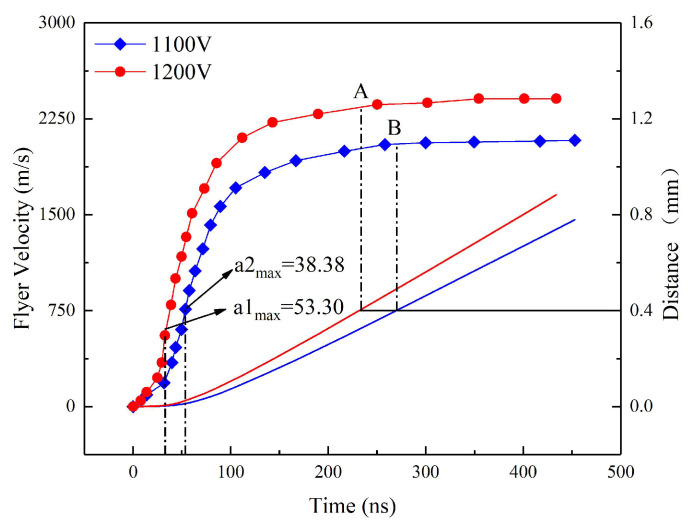
Velocity-time and distance-time curves of the flyer at 1100 V/0.22 μF and 1200 V/0.22 μF.

**Table 1 micromachines-11-00550-t001:** Short-circuit discharge parameters at corresponding voltages.

Voltage /V	*I*_peak1_ /A	*I*_peak2_ /A	*T* /ns	*R*_0_ /mΩ	*L* /nH
600	1172.33	440.858	470	103.3	24.83
800	1651.31	773.72	447.8	77.1	22.76
1000	2170.54	1170.21	426	60.0	20.69
1200	2619.52	1555.44	418	49.8	19.98

**Table 2 micromachines-11-00550-t002:** Peak voltage of the G-GR at different capacitor voltages.

Capacitor Voltage/V	Gate Voltage/V
600	13.67
800	14.25
1000	14.66
1200	14.69

**Table 3 micromachines-11-00550-t003:** Electric explosion parameters at different voltages.

Voltage /V	*I*_peak_ /kA	*t*-*I*_peak_ /ns	*V*_peak_ /kV	*t*-*V*_peak_ /ns	*P*_peak_ /MW	*t*-*P*_peak_ /ns	*E_d_* /mJ
600	0.97	202.98	0.11	139.58	0.08	160.58	4.12
800	1.35	192.09	0.16	121.49	0.18	147.69	9.22
1000	1.68	171.35	0.47	238.35	0.40	229.35	45.39
1200	1.96	143.96	0.72	191.96	0.90	187.16	63.36

**Table 4 micromachines-11-00550-t004:** Initiation results of two kinds of HNS.

Particle Size (D50) /nm	Capacitor Voltage /kV	Capacitor Energy /mJ	Initiation or Not
373	1.30	185.9	yes
373	1.20	158.4	yes
373	1.10	133.1	no
158	1.20	158.4	yes
158	1.10	133.1	yes
158	1.00	110.0	no

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
