# Peer review of "Firing Performance of Microchip Exploding Foil Initiator Triggered by Metal-Oxide-Semiconductor Controlled Thyristor"

_micromachines, 2020, doi:10.3390/mi11060550_

Round 1
Reviewer 1 Report
The authors present an interesting study of an exploding foil initiator that can be triggered by an MOS controlled thyristor. The engineering aspects of the paper are mostly satisfactory but the scientific components are lacking. Here are some detailed comments.
- My biggest complaint about the paper is that it is not very clear what the improvement is over the literature. It seems to be that the MCT technology has not often been applied to EFI detonators, and I am guessing that the only reason it works here is that HNS is relatively sensitive to flyer impact so you don't need so high of a voltage. Is that right? Presenting that very clearly in the abstract would be helpful. If MCT has ever been studied with EFI that needs to be clearly cited with appropriate references instead of just saying "few public reports..." As far as I can tell, using MCT instead of another high voltage switch is really the only novel part of the paper. If the authors believe this is very important and a great contribution to the field, they should emphasize this as much as possible. And why is MCT preferable to commercially available gas switches (or others)? I got some idea from the introduction but the only downside mentioned to commercial is price... Can you also add WHY MCT switches are not commonly used?
- The EFI design is not groundbreaking and in fact looks very much like that of Fang Kuang et al (Fang Kuang et al. Chinese Journal of Energetic Materials, 24 (9), 2016, 892-897.) They also used MEMS techniques to fabricate EFI, and I believe that MEMS techniques are very common in the field anyways. That paper (and maybe others) should absolutely be cited and any differences in EFI build details (e.g. flyer thickness) should be called out and if they are improvements.
- Multiple acronyms are not defined the first time they are used. Please correct this. I noticed McEFI and MCT were not defined - there may be others. I noticed some (like EFI) were defined multiple times.
- HNS-IV needs a definition. How was the HNS synthesized? How was the ball-milling performed? How were particle sizes measured, and is there a range of sizes or are they all perfectly identical?
- How was HNS detonation assessed? There can be multiple types of violent reactions but "detonation" is very specific.
- How many tests were performed at each condition in Table 4 to determine yes or no detonation?
- Some context would be useful for the reader to interpret how much of an improvement is made with the MCT compared to the literature, especially reference 7. The capacitor voltage here was up to 300 V lower and the energy difference is less than 100 mJ. Is that really much of an improvement? How important is this?
- It seems like the "Conclusions" section is mostly about MCT advantages that could answer my first question and also be placed in the Introduction. Conclusions should be only about what was proved in this present work, of which only the first sentence is addressing. The last two sentences are fine too since they point to limitations of the present work and possible future work.
- In general, experimental error is not addressed. There are no reported measurements of the McEFI dimensions - is the flyer thickness really what you targeted, or are there some small thickness variations from the deposition? There are no error bars or discussion of error from the measurement technique for current or voltage as a function of time.
Reviewer 2 Report
The work by Wang et al. (Firing Performance of Microchip Exploding Foil Initiator Triggered by MOS Controlled Thyristor) reports the electric explosion performance of microchip exploding foil initiators (McEFI) through a small-scale capacitor discharge unit based on MOS controlled thyristor. For the fabrication of McEFI, they employed some micro-electro-mechanical system scale fabrication methods. The detonation ability of McEFI was tested for two HNS-IV for two different particle size. It was shown that both voltage and particle size have significant effect on detonation of the proposed system.
The manuscript is well-organised and easy-to-follow. I recommend the publication of the report after the some issues given below are addressed.
Please define the open name of the MOS in the title and abstract when it is first mentioned.
Line 73
Please use only the abbrevation.
……. microelectromechanical system (MEMS)….
A scale bar can be added to Figure 1 to show the size of the McEFI. Also the cross-section of the final system would be informative to figure out the structure of the system.
An optic image with higher magnification can be added to Figure 2 as an inset.
Round 2
Reviewer 1 Report
Thank you to the authors for thoughtful responses and reworking of the manuscript. The purpose and key conclusions of the paper have been sufficiently clarified. I think the reader would still benefit from more information in the manuscript about how initiation was assessed - the wording used in your answer to me about visual observation (PCB board destroyed, smoke, etc) would be helpful to have in the manuscript. If the reader is unconvinced initiation occurred due to having no details, they are less likely to consider and use your MCT-based design. Otherwise the paper has been improved enough for publishing in my opinion.
Author Response
Dear reviewer:
Thank you very much for your quickly comments about our paper submitted to Micromachines again. (Manuscript ID: 803535, Title: Firing performance of Microchip Exploding Foil Initiator Triggered by Metal-Oxide-Semiconductor Controlled Thyristor). Thank you for confirming my last response.
According to your advice, I have added the corresponding content in the article in correspondence with your requirements.
Comment 1:
Thank you to the authors for thoughtful responses and reworking of the manuscript. The purpose and key conclusions of the paper have been sufficiently clarified. I think the reader would still benefit from more information in the manuscript about how initiation was assessed - the wording used in your answer to me about visual observation (PCB board destroyed, smoke, etc) would be helpful to have in the manuscript. If the reader is unconvinced initiation occurred due to having no details, they are less likely to consider and use your MCT-based design. Otherwise the paper has been improved enough for publishing in my opinion
Answer 1:
The related change can be found in Line 291-294.
